# Management and Outcomes of Pancreatic Cancer in French Real-World Clinical Practice

**DOI:** 10.3390/cancers14071675

**Published:** 2022-03-25

**Authors:** Valérie Jooste, Leila Bengrine-Lefevre, Sylvain Manfredi, Valérie Quipourt, Pascale Grosclaude, Olivier Facy, Côme Lepage, François Ghiringhelli, Anne-Marie Bouvier

**Affiliations:** 1Digestive Cancer Registry of Burgundy, Dijon University Hospital, INSERM UMR 1231 EPICAD, Medical School, University of Burgundy-Franche Comté, 21000 Dijon, France; valerie.jooste@u-bourgogne.fr; 2Department of Medical Oncology, Georges-Francois Leclerc Cancer Center, Geriatric Oncology Coordination Unit in Burgundy, University Hospital, 21000 Dijon, France; lbengrine@cgfl.fr; 3University Hospital of Dijon, CRCDC BFC (Centre Régional de Coordination de Dépistage des Cancers Bourgogne Franche Comté), UMR INSERM 2131 EPICAD, Medical School, University of Burgundy-Franche Comté, 21000 Dijon, France; sylvain.manfredi@chu-dijon.fr; 4Department of Geriatrics and Internal Medicine, Hospital of Champmaillot, Geriatric Oncology Coordination Unit in Burgundy, University Hospital, 21000 Dijon, France; valerie.quipourt@chu-dijon.fr; 5Institut Claudius Regaud, IUCT-O, Registre des Cancers du Tarn, University of Toulouse Paul Sabatier, INSERM 1027, 31059 Toulouse, France; pascale.grosclaude@inserm.fr; 6Department of Digestive Surgery, Dijon University Hospital, INSERM UMR 1231, Medical School, University of Burgundy-Franche Comté, 21000 Dijon, France; olivier.facy@chu-dijon.fr; 7Department of Hepatogastroenterology and Digestive Oncology, University Hospital of Dijon, INSERM UMR 1231 EPICAD, Medical School, University of Burgundy-Franche Comté, 21000 Dijon, France; come.lepage@u-bourgogne.fr; 8Department of Medical Oncology, Georges François Leclerc Cancer Center-UNICANCER, UMR INSERM 1231, Medical School, University of Burgundy-Franche Comté, 21000 Dijon, France; fghiringhelli@cgfl.fr

**Keywords:** pancreatic cancer, primary care, epidemiology, cancer registries, chemotherapy

## Abstract

**Simple Summary:**

Surgical resection is the only potentially curative treatment for pancreatic cancer, and its indication relies on precise imaging criteria and on patients’ operability. Chemotherapy is recommended, except for patients a with very short life expectancy. We sought to provide real-world information on the management and outcomes of pancreatic cancer. Among patients with a surgically resectable tumor, only half of those aged 75–84 and none after 85 actually underwent resection, even though the prognosis following pancreatectomy in elderly patients was similar to that in younger patients. Patients’ refusal of chemotherapy increased from 7% before 75 years to 73% after 85 years. These results underline the need to develop guidelines for the management of elderly patients with pancreatic cancer and to generalize geriatric assessments.

**Abstract:**

Background: Our objective was to describe real-world patterns of care and outcomes in pancreatic cancer. Methods: 912 patients diagnosed with pancreatic cancer from 2014 to 2017 were registered by the population-based cancer registry of Burgundy (France). Progression-free and net survival were estimated. Results: at diagnosis, 52% of tumors were associated with metastases. Among the 20% of patients fulfilling resectability criteria, half of those aged 75–84 years and none of those ≥85 years actually underwent resection. Age was not associated with 3-year observed survival in patients who underwent resection. Overall, 77% of patients aged <75 years, 55% of those aged 75–84 years and 8% of those ≥85 years received chemotherapy. Among patients who were offered chemotherapy, 73% of those aged ≥85 years refused. Chemotherapy toxicity was higher with Gemcitabine_Oxaliplatin/Gemcitabine_Abraxane and FOLFIRINOX than with Gemcitabine alone. Patients resected after induction FOLFIRINOX and those treated with adjuvant Gemcitabine presented the lowest risk of progression. Three-year net survival was 35% in patients with non-metastatic resectable tumors and under 10% for other patients. Conclusions: Only half of patients aged 75–84 years with a resectable tumor actually underwent resection. Two thirds of patients aged ≥85 years refused chemotherapy, thus underlining the need to expand geriatric assessments.

## 1. Introduction

In recent decades, pancreatic cancer has become a major public health concern, ranking as the 7th most frequent cancer and the 4th most common cause of cancer-related death in the European Union [1,2]. Though surgical resection remains the only potentially curative treatment, few patients are deemed suitable for the treatment. In clinical trials, around 70–80% of patients relapsed after tumor resection within a median time of around 12 months [3]. In the 1990s and the 2000s, the European Society for Medical Oncology (ESMO) guidelines recommended adjuvant chemotherapy, based on Gemcitabine, then on the association of Gemcitabine and Capecitabine, and more recently on FOLFIRINOX [4]. It has been suggested that induction chemotherapy can improve resectability through tumor reduction, and thereby reduce the morbidity of pancreatic surgery, even though no phase III trials have supported this hypothesis. For patients unfit for surgical resection, palliative systemic chemotherapy using either FOLFIRINOX or gemcitabine, alone or combined with nab-paclitaxel, has shown a survival benefit [5].

Data provided by specialized centers or clinical trials cannot be used to describe patterns of care because of unavoidable selection bias. Population-based studies that include all cases diagnosed in a well-defined population are the best way to picture the management of cancer in the real-world. Such studies are rare because they require accurate and detailed data collection through dedicated surveys on representative samples of cases.

Thus, the objective of this study was to describe the real-world management and outcomes of pancreatic cancer in a population-based cohort study.

## 2. Materials and Methods

### 2.1. Patients

The population-based digestive cancer registry includes all digestive cancers diagnosed in the inhabitants of two administrative areas in France (counties of Côte-d’Or and Saône-et-Loire, Burgundy, 1,082,000 inhabitants). The quality and comprehensiveness of the registry is certified every four years by an audit of the National Public Health Institute (Santé Publique France, Paris, France), the French National Cancer Institute (INCa, Institut national du cancer, Paris, France), and the National Institute for Health and Medical Research (INSERM). Information is collected from pathology laboratories, university and local hospitals, private physicians (surgeons, gastroenterologists, oncologists, and general practitioners), social security offices, and death certificates. No case is registered through the death certificate alone, and all death certificates mentioning digestive cancer are individually tracked. This observational non-interventional study was approved by the French Data Protection Authority (CNIL, authorization n° 998024), and in agreement with French legislation, there was no requirement for written informed consent.

All malignant exocrine pancreatic tumors (coded as C25 in accordance with the International Classification of Diseases in Oncology, third revision) diagnosed between 2014 and 2017 in residents ≥18 years were extracted from the registry database. Patients with benign/premalignant tumors, neuroendocrine tumors, stromal tumors, sarcoma, lymphoma or peri-ampullar cancers were excluded. This investigation was conducted according to the Declaration of Helsinki and the STROBE guidelines.

### 2.2. Data Set

Information about age, sex, clinical features, metastatic status at diagnosis, and treatment modalities (surgical resection, induction, adjuvant or palliative chemotherapy, and radiotherapy) was routinely collected. The ECOG Performance Status at diagnosis (before any treatment) and the Charlson comorbidity index were categorized into the following groups: “0–1”, “2”, and “3+”. Performance status was missing for 78 cases and the Charlson index for 11 cases. Socio-economic status was assessed at the level of the “IRIS”, which are the smallest geographic areas defined by the ‘Institut National de la Statistique et des Etudes Economiques’ for which census data are available [6]. Each IRIS includes approximately 2000 individuals with relatively homogeneous social characteristics. An IRIS was assigned to each patient according to their residence address at the time of diagnosis. The French version of the ecological European Deprivation Index (EDI), an aggregate index of deprivation based on the 2011 national census, was used to assign an EDI score to each IRIS [7]. The continuous EDI was categorized into national quintiles (the higher the quintile, the greater the social deprivation). The EDI was missing for 13 cases.

Through a dedicated survey in all concerned health structures, detailed complementary information was collected from individual medical files and from multidisciplinary team meetings. For patients with non-metastatic disease, the surgical resectability criteria (arterial and venal invasion, regional lymph node extension, extra-pancreatic local extension) were recorded from computed tomography scans (CT-scan) [8]. According to the clinical and morphological examination reported in the medical files, patients were categorized into three clinical features categories: (1) non-metastatic disease (M0) with a resectable tumor, (2) non-metastatic disease with a locally advanced unresectable tumor, and (3) initial distant metastasis (M1). Initially, unresectable tumors rendered resectable by induction chemotherapy were classified as resectable. Clinical features were missing for 39 cases.

A patient was considered to have received a first line of chemotherapy if at least one cycle was administered, regardless of the dose or administration method. Information on chemotherapy administration was available in 903 patients. The first-line treatment scheme (missing for two patients) was classified as:(0)No chemotherapy (*n* = 375)(1)gemcitabine without resection (Gem alone, *n* = 92),(2)gemcitabine + oxaliplatin or gemcitabine + nab-paclitaxel without surgical resection (Gem_Ox/Gem_Abra alone, *n* = 78); these were pooled as patients presented similar clinical features (mostly locally advanced or M1 disease) and similar characteristics,(3)leucovorine + 5-fluorouracil + irinotecan + oxaliplatin without surgical resection (FOLFIRINOX: FFX alone, *n* = 255),(4)FOLFIRINOX followed by surgical resection (FFX induction, *n* = 14),(5)gemcitabine (*n* = 63) or gemcitabine + capecitabine (*n* = 24) after surgical resection (Gem adjuvant, *n* = 87). In accordance with ASCO guidelines [9] stating that doublet therapy with gemcitabine and capecitabine or mono-therapy with gemcitabine alone can be offered in adjuvant settings; gemcitabine + capecitabine and gemcitabine alone were pooled.

FOLFIRINOX after surgical resection became the standard after the period of this study [4,9]. The chemotherapy scheme was unknown in two cases. Reasons for non-administration were collected in a dedicated survey, and included a medical contra-indication, an operative complication, age, performance status, early death, or refusal from the patient or family. It was unknown for 40 patients. Due to the population-based observational design, toxicity grades were not always reported in medical files according to the World Health Organization classification. Thus, we classified chemotherapy toxicity as: no toxicity, toxicity with no consequences on the chemotherapy regimen (mild), and toxicity leading to a change in the chemotherapy regimen or its interruption (severe). Toxicities were described as Neurological toxicity, Hematological toxicity, Digestive toxicity, or General toxicity (asthenia, allergy, headache, renal failure). Disease progression was assessed and characterized in patients with non-metastatic disease undergoing surgical resection or receiving chemotherapy. Progression was assessed according to the RECIST (Response Evaluation Criteria in Solid Tumors) definition. The registry staff retrospectively collected and dated the iterative RECIST conclusions evaluated by the patients’ practitioners. Progression was available in 91% of patients with non-metastatic cancer who underwent surgical resection or first-line chemotherapy. The vital status of patients was ascertained through an electronic request to the National directory for the identification of persons or from the register of the place of birth or residence. Vital status at 1 January 2019 was known for 98.1% of cases (*n* = 895).

### 2.3. Statistical Analysis

The association between categorical data was analyzed using the chi-square test (or Fisher’s exact test if needed). Incidence rates were standardized by the direct method, using the world standard population. Age was divided into 18–64, 65–74, 75–84, and ≥85 years. The EDI deprivation index is a continuous variable. All the French IRIS are distributed in quintiles of deprivation, according to their EDI value, over the national territory. Quintile 1 contains the least deprived IRIS and Quintile 5 contains the most deprived.

Non-conditional multivariate logistic regression was used to identify factors independently associated with the probability of having surgical resection, chemotherapy, and presenting toxicity. The significance of the covariates was tested by the likelihood ratio test.

The cumulative probability of progression free survival (PFS) was estimated by the Kaplan–Meier method, considering the studied event as the first event between RECIST progression and all-cause death. Patients alive and free of RECIST progression were censored at the end of follow-up. In order to limit immortal time bias, surgical resection alone (*n* = 23) was pooled with adjuvant Gemcitabine (*n* = 87).

Net survival represents the survival of patients in the hypothetical situation in which cancer is the only cause of death [10] and may be interpreted as cancer survival after controlling for competing causes of death. It can be estimated using the excess mortality method, based on the difference between the mortality observed in the studied cohort and the expected mortality. For each sex and administrative area, these expected mortality rates were derived from the general population mortality rates as provided by the Institut National de la Statistique et des Études Économiques, smoothed by the Service de Biostatistique-Bioinformatique des Hospices Civils de Lyon. Net survival was estimated on the whole patient cohort using the flexible parametric model proposed by Nelson et al. [11].

Observed survival (OS) was estimated using the Kaplan–Meier univariate method and Cox multivariate model and added as a means to draw parallels between our results and those of published randomized trials.

Stata (Stata Statistical Software: Release 17. College Station, TX: StataCorp LLC) was used for all analyses. Statistical significance was defined by two-sided *p* < 0.05.

## 3. Results

Over the period 2014–2017, 912 pancreatic cancers fulfilling the inclusion criteria were recorded (481 men and 431 women). The corresponding world standardized incidence was 10.7 per 100,000 in men and 7.5 per 100,000 in women. Mean age at diagnosis was 70.3 (SE: 11.4) for men and 76.3 (SE: 11.3) for women (*p* < 0.001) (Table 1). The proportion of patients with surgically resectable tumors, as defined during multidisciplinary team meetings, decreased strongly after 75 years (15% vs. 25% before 75 years), whereas the proportions of patients with locally advanced (30% vs. 21%) or metastatic disease (54% vs. 55%) was similar for both categories of age.

### 3.1. Surgical Resection

Initially, 175 patients with non-metastatic disease presented with a resectable tumor. Additionally, five tumors initially classified as unresectable were rendered resectable after induction chemotherapy. The reasons for non-resectability were arterial and/or venal invasion in 80% of cases, and/or regional lymph node extension in 27% of cases, and/or extra-pancreatic local extension in 24% of cases. Among the 180 patients with tumors fulfilling the resectability criteria, 126 actually underwent resection (105 pancreato-duodenectomies, 19 distal pancreatectomies, and 2 total pancreatectomies). Almost 90% of patients aged <75 years with resectable tumors underwent resection, while half of those aged 75–84 and none of those ≥85 years did (Table 2). Age, sex, and performance status were significantly associated with the likelihood of tumor resection, whereas the Charlson index, deprivation, and tumor location were not. After adjustment for age, patients with a performance status >1 were far less likely to undergo resection than patients with a performance status = 0–1 (*p* < 0.001). The association between sex and resection was no longer significant. Concerning 30-day postoperative mortality, 4/104 patients aged 15–74 and 0/22 patients aged 75–84 died. Corresponding figures for 90-day postoperative mortality were 8/104 and 1/22.

### 3.2. Chemotherapy

Overall, 28% of patients were not offered chemotherapy. This was related to performance status and old age in 74%, to early death in 14% and to a contra-indication in 9%. Among the 615 patients who were offered chemotherapy, 14% refused (refusal expressed by patients or their families). This proportion increased from 7% before 74 years, to 16% between 75 and 84 years, and 73% after 84 years (*p* < 0.001). The likelihood of refusal was higher in patients with a performance status > 1 (31%) than in those with a performance status = 0–1 (9%, *p* < 0.001), whereas the Charlson index was not associated with refusal.

Among the 912 patients, 528 actually received at least one line of chemotherapy (80% palliative, 17% adjuvant, 3% induction). In univariate analysis, 77% of patients <75 years received chemotherapy, 55% of those aged 75–84 and 8% of those >84 (Table 3). Fifty-nine percent of patients with locally advanced disease received chemotherapy, as did 56% of patients with metastatic disease and 69% of patients with surgically resectable tumors (*p* = 0.008). Of these, respectively 18%, 1%, and 2% also received radiotherapy. Among the 126 patients who underwent resection, 90 (71%) received adjuvant chemotherapy and 15 (12%) received induction chemotherapy, whereas among the 54 patients with resectable tumors who could not undergo surgical resection, 21 received chemotherapy. In multivariate analysis, older age, a Charlson index ≥3, and a performance status >1 were associated with not receiving chemotherapy, whereas the clinical features were not (Table 3). The deprivation quintile was associated with chemotherapy administration in both univariate and multivariate analyses: the higher the deprivation quintile, the lower the likelihood of receiving chemotherapy. When the deprivation index was included as a continuous variable, the linear trend was non-significant in multivariate analysis (OR = 0.96 [0.92–1.01], *p* = 0.099).

### 3.3. Chemotherapy-Related Toxicities

Thirty-one percent of treated patients aged < 85 years presented at least one severe toxicity, compared with 67% of patients over 85 years (*p* = 0.017, Table 4). FOLFIRINOX was associated with toxicity in 40% of cases, Gemcitabine + Oxaliplatin/Gemcitabine + Abraxane in 33%, Gemcitabine alone in 26%, and adjuvant Gemcitabine in 14% (*p* < 0.001). Toxicity was not associated with the Charlson index or performance status. In multivariate analysis, toxicity increased with age (*p* < 0.001) and varied with chemotherapy regimen. Patients receiving FOLFIRINOX were four times as likely and those receiving Gemcitabine + Oxaliplatin/Gemcitabine + Abraxane were twice as likely to experience toxicity, as were patients receiving Gemcitabine alone or adjuvant Gemcitabine. Additionally, the proportion of patients presenting mild toxicities was 23% for FOLFIRINOX, 17% for Gemcitabine + Oxaliplatin/Gemcitabine + Abraxane, 20% for Gemcitabine alone, and 22% for adjuvant Gemcitabine (Appendix A). Among patients who received FOLFIRINOX, 22% presented neurological toxicities (9% mild and 13% severe), 21% digestive toxicities (10% mild and 11% severe), 18% hematological toxicities (6% mild and 12% severe), 25% general toxicities (4% mild and 21% severe), and 37% no toxicity.

### 3.4. Progression Free Survival (PFS)

Among the 287 patients at risk of disease progression, overall, 6-month, 1-, 2-, and 3-year PFS was 80%, 51%, 23%, and 13%, respectively. Patients who underwent tumor resection with or without adjuvant Gemcitabine and those who had tumor resection after induction FOLFIRINOX presented the lowest risk of progression, with a 2-year PFS of 38% and 25%, respectively (Figure 1). Median PFS was 12 months in patients eligible for treatment with Gemcitabine + Oxaliplatin/Gemcitabine + Abraxane or with FOLFIRINOX alone, and 7 months after Gemcitabine alone.

### 3.5. Survival

Six-month, 1-year, and 3-year net survival was 56%, 37%, and 11%, respectively. In univariate analysis, age, the Charlson index, performance status, deprivation, and clinical features were significant prognostic factors, while sex was not (Table 5). Three-year net survival was 35% in patients with non-metastatic resectable tumors compared with less than 10% in patients with non-resectable tumors or metastatic disease. After adjustment for age and clinical features, a Charlson index of 3+ and any deterioration in performance status were associated with a poor prognosis.

Median observed survival (OS) was longer than 2 years in patients with resected tumors who received chemotherapy: it was 30 months (95% CI (24–36)) in patients able to receive adjuvant Gemcitabine and 24 months (13–36) in those able to receive induction FOLFIRINOX. It was 17 months (10–24) in patients undergoing resection alone. In the absence of resection, median OS was shorter than 1 year: it was 6 months (4–7) in patients receiving palliative gemcitabine alone and similar in those receiving palliative Gemcitabine + Abraxane (11 months (8–13)) or palliative FOLFIRINOX (12 months (10–14)). It made no sense to statistically compare regimens due to indication bias.

In M0 patients who underwent resection, age was not associated with 3-year OS in univariate (*p* = 0.722, Figure 2) or multivariate analysis: HR _65–74_ vs. _<65_ = 1.12 (0.68–1.87) and HR _75–84_ vs. _<65_ = 0.76 (0.38–1.55) (*p* = 0.501 data not shown).

## 4. Discussion

This population-based study showed that in patients with a resectable tumor, only half of those aged 75–84 years and none of those older than 85 actually underwent resection. The prognosis following pancreatectomy in elderly patients was nevertheless similar to that in younger patients. Chemotherapy was refused by three-quarters of patients older than 85 to whom it was offered.

To our knowledge, this is the first detailed European population-based study on resectability, chemotherapy regimens and related toxicities, disease progression and survival in patients with pancreatic cancer.

The small proportion (14%) of resected cases in our study was similar to that in other European or American countries, ranging from 12 to 22% [12,13,14,15], The centralization of pancreatic surgery could increase the proportion of patients undergoing resection for pancreatic cancer [16]. Performance status was related to the likelihood of surgery, while the Charlson Comorbidities index was not. The Charlson index was associated with the likelihood of surgery in an Australian population-based study [17] whereas it was not in an American SEER-Medicare analysis [18]. This inconsistency in large cohorts may partly result from heterogeneity in the source of information (medical file, administrative database…) and its reporting (medical staff, technicians, algorithms…). Information on performance status seems better able than the Charlson index to reflect the suitability of patients for surgery and burdensome treatments.

Given the high morbidity of pancreatectomy, the importance of age in the surgical decision is not surprising. The fact that postoperative mortality and 3-year net survival following pancreatectomy are similar in all age groups suggests that the prognosis of pancreatic cancer in the elderly could be improved by selecting more patients for surgery. The spread of geriatric assessments could help decision-making in multidisciplinary team meetings and lead to a wider selection of older cancer patients fit for surgery or chemotherapy [19]. Although there was no difference between age groups for the proportion of locally advanced or metastatic tumors, chemotherapy was administered to three-quarters of those aged less than 75 compared with half of those aged 75–84 years. Three quarters of patients diagnosed after 85 years refused chemotherapy, whereas refusal was rare in younger patients. These findings reflect how patients are treated in everyday practice in a country where chemotherapy in public and private practice is provided free of charge to patients. This high refusal rate in the elderly could be related to the conjunction of low benefit and high toxicity, which could make clinicians less keen to use chemotherapy. Even though little is known about factors used to select patients for treatment, the 75 years of age threshold for FOLFIRINOX administration, which is usual in clinical trials, was actually observed in real world clinical practice.

After chemotherapy, severe toxicity occurred in one third of patients. Although FOLFIRINOX was administered to younger patients, there were four times more toxicities with this combination than with gemcitabine. However, in our study, FOLFIRINOX and gemcitabine combinations provided similar progression-free and net survival. Results from observational studies have to be assessed with caution because of the absence of randomization. Nevertheless, they may be useful to evaluate the external validity of the results of clinical trials [20]. Our findings strengthen the results from two randomized clinical trials [5,21] in which gemcitabine/nab-paclitaxel in one and FOLFIRINOX in the other as compared to gemcitabine alone improved survival. Modified FOLFIRINOX schemes could be proposed to decrease toxicities in the future.

This population-based study has the advantage of including detailed descriptions of all patients with pancreatic cancer diagnosed in a population of more than one million inhabitants. It thus avoids the referral bias that often occurs in hospital-based series. The high quality and completeness of active clinical follow-up and the multi-institutional nature of the study population strengthen the reliability and highlight the wide scope of our results.

However, the study was limited by its retrospective observational nature. The generalizability of our findings, particularly with respect to clinical practices, depends as much on the system of care as on the modalities of its use, which may vary with the social characteristics of patients [22]. However, the supply of healthcare in the studied region showed similar characteristics to those of other French geographic areas (reference academic cancer centers, private and public hospitals). Access to radiotherapy, surgery, or chemotherapy in both private and public structures was unlimited, and the rules for authorizing the activities of healthcare facilities have been defined for the whole of France by the national Cancer Plan (https://www.e-cancer.fr/Professionnels-de-sante/L-organisation-de-l-offre-de-soins/Traitements-du-cancer-les-etablissements-autorises/Les-autorisations-de-traitement-du-cancer (accessed on 15 Ferbruary 2022)). French patients have the same access to national health insurance-funded care and to any place of treatment, whatever its geographic situation.

Indication bias could have led to differences in patients’ characteristics according to treatment modality. Potential immortal time bias was eliminated by pooling patients who underwent surgical resection alone with patients who received adjuvant chemotherapy. The registration of progression relied on RECIST imaging criteria, but clinical progression was not considered. One limit of the study was its relatively small size in some treatment categories according to age. We cannot exclude the possibility that some differences may become significant with a larger number of patients, but none of the reported differences were close to the significance threshold (5%). Because of the unavoidable delay in recording the data in a population-based registry, the most recent chemotherapy regimen patterns, such as adjuvant FOLFIRINOX [9], could not be considered.

## 5. Conclusions

Our results suggest that the generalization of geriatric assessments could improve surgical and medical care of elderly patients with pancreatic cancer and should be further assessed in epidemiological studies.

## Figures and Tables

**Figure 1 cancers-14-01675-f001:**
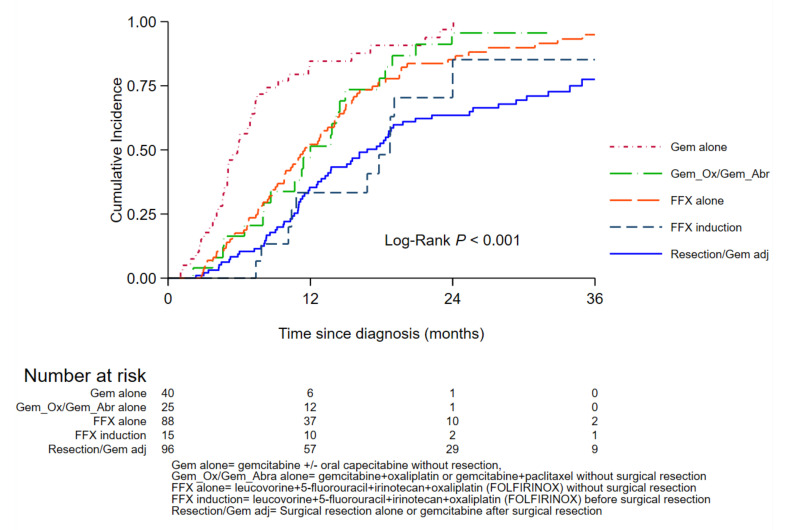
Three-year cumulative probability of progression or death according to treatment modalities (surgical resection and first-line chemotherapy regimen).

**Figure 2 cancers-14-01675-f002:**
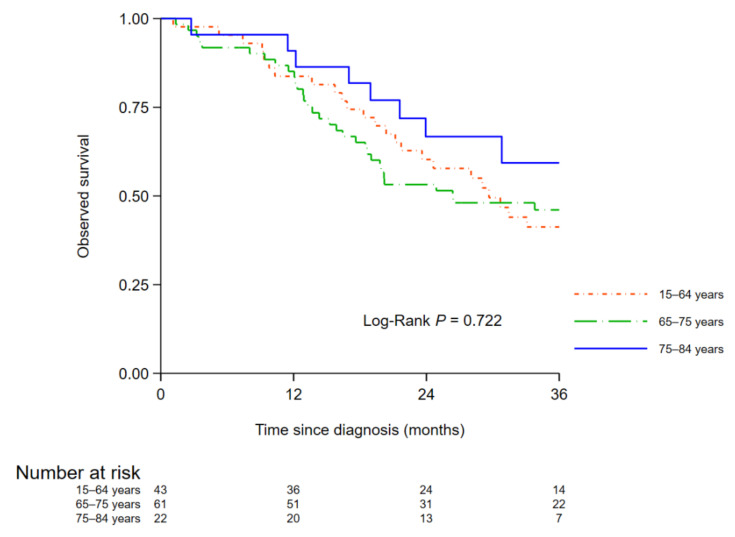
Three-year observed survival among patients with resected non-metastatic pancreatic cancer by age.

**Table 1 cancers-14-01675-t001:** Description of the study population according to age at diagnosis of pancreatic cancer (*n* = 912).

	Total	<65 y *n* (%)	65–74 y *n* (%)	75–84 y *n* (%)	≥85 y *n* (%)	*p* ^a^
Sex						
Men	481	152 (68)	160 (60)	114 (45)	55 (33)	
Women	431	73 (32)	108 (40)	139 (55)	111 (67)	<0.001
Charlson index						
0–1	660	184 (82)	192 (73)	170 (68)	114 (70)	
2	146	28 (13)	41 (16)	49 (20)	28 (17)	
3+	95	12 (5)	31 (12)	32 (13)	20 (12)	0.019
Performance Status						
0–1	588	179 (83)	207 (82)	150 (66)	52 (38)	
2	174	29 (13)	36 (14)	57 (25)	52 (38)	
3–4	72	7 (3)	10 (4)	22 (10)	33 (24)	<0.001
Deprivation						
Quintile 1	168	38 (17)	61 (23)	48 (19)	21 (13)	
Quintile 2	171	48 (22)	57 (22)	40 (16)	26 (16)	
Quintile 3	154	37 (17)	43 (16)	39 (16)	35 (21)	
Quintile 4	227	58 (26)	60 (23)	66 (26)	43 (26)	
Quintile 5	179	41 (18)	41 (16)	58 (23)	39 (24)	0.104
Clinical features ^b^						
M0 resectable tumor	180	49 (22)	70 (27)	43 (18)	18 (12)	
Locally advanced	217	46 (21)	53 (21)	60 (25)	58 (38)	
M1	476	127 (57)	135 (52)	137 (57)	77 (50)	<0.001
Treatment						
Surgical resection without chemotherapy	21	6 (3)	12 (5)	3 (1)	0 (0)	
Best supportive care	354	43 (19)	49 (18)	109 (44)	153 (93)	
Chemotherapy	528	173 (78)	204 (77)	139 (55)	12 (7)	<0.001
*1^st^ line regimen* ^c^ *:*						
*Gem alone*	*92*	*8 (4)*	*22 (8)*	*51 (20)*	*11 (7)*	
*Gem_Ox/Gem_Abra alone*	*78*	*13 (6)*	*26 (10)*	*38 (15)*	*1 (1)*	
*FFX alone*	*255*	*116 (52)*	*108 (41)*	*31 (12)*	*0 (0)*	
*FFX induction*	*14*	*9 (4)*	*5 (2)*	*0 (0)*	*0 (0)*	
*Gem adjuvant*	*87*	*26 (13)*	*42 (16)*	*19 (8)*	*0 (0)*	*<0.001*

^a^ Chi-square test. ^b^ M0 = non-metastatic, M1 = metastatic. ^c^ Gem alone = gemcitabine +/- oral capecitabine without surgical resection. Gem_Ox/Gem_Abra alone = gemcitabine + oxaliplatin (*n* = 46) or gemcitabine + nab-paclitaxel (*n* = 32) without surgical resection. FFX alone = leucovorine + 5-fluorouracil + irinotecan + oxaliplatin (FOLFIRINOX) without surgical resection. FFX induction = leucovorine + 5-fluorouracil + irinotecan + oxaliplatin (FOLFIRINOX) followed by surgical resection. Gem adjuvant = gemcitabine after surgical resection.

**Table 2 cancers-14-01675-t002:** Factors associated with the occurrence of surgical resection in patients with resectable M0 pancreatic cancer (*n* = 180).

		Surgical Resection		Multivariate Analysis ^c^		
	Total	*n* (%)	*p* ^a^	AOR	(95%CI)	*p*
Age						
<65 years	49	43 (88)		1		
65–74 years	70	61 (87)		1.58	(0.47–5.35)	
75–84 years	43	22 (51)		0.22	(0.07–0.66)	<0.001
≥85 years	18	0 (0)	<0.001	-		
Sex						
Men	97	75 (77)		1		
Women	83	51 (61)	0.021	0.58	(0.24–1.39)	0.221
Charlson index						
0–1	125	88 (70)		-		
2	31	23 (74)		-		
3+	21	12 (57)	0.390	-		
Deprivation						
Quintile 1	29	19 (66)		-		
Quintile 2	43	30 (70)		-		
Quintile 3	27	18 (67)		-		
Quintile 4	54	39 (72)		-		
Quintile 5	24	17 (71)	0.971	-		
Performance Status						
0–1	152	118 (78)		1		
2	17	3 (18)		0.08	(0.02–0.40)	
3–4	8	2 (25)	<0.001 ^b^	0.10	(0.01–0.93)	<0.001
Location						
Head	146	102 (70)		-		
Other	34	24 (71)	0.934	-		

^a^ Chi-square test except for ^b^ Fisher’s exact test. ^c^ Logistic multivariate regression model: 18 patients >85 years excluded, *p*: likelihood ratio test, AOR: Adjusted Odds Ratio.

**Table 3 cancers-14-01675-t003:** Administration of chemotherapy in patients with pancreatic cancer (*n* = 912).

		Chemotherapy		Multivariate Analysis ^c^		
	Total	*n* (%)	*p* ^a^	AOR	(95%CI)	*p*
Age						
<65 years	222	173 (78)		1		
65–74 years	265	204 (77)		1.10	(0.67–1.81)	
75–84 years	251	139 (55)		0.46	(0.29–0.75)	
≥85 years	165	12 (7)	<0.001	0.03	(0.01–0.07)	<0.001
Sex						
Men	475	292 (61)		1		
Women	428	236 (55)	0.054	1.38	(0.94–2.03)	0.103
Charlson index						
0–1	657	409 (62)		1		
2	145	81 (56)		1.27	(0.75–2.13)	
3+	95	36 (38)	<0.001	0.47	(0.26–0.84)	0.016
Performance Status						
0–1	585	440 (75)		1		
2	174	64 (37)		0.30	(0.19–0.46)	
3–4	72	8 (11)	<0.001	0.07	(0.03–0.15)	<0.001
Deprivation						
Quintile 1	168	105 (63)		1	(0.80–2.72)	
Quintile 2	170	119 (70)		1.47	(0.80–2.72)	
Quintile 3	154	88 (57)		1.02	(0.55–1.89)	
Quintile 4	221	124 (56)		0.79	(0.46–1.38)	
Quintile 5	177	83 (47)	<0.001	0.59	(0.33–1.05)	0.033
Clinical Feature ^b^						
M0 resectable tumor	176	122 (69)		1		
Locally advanced	216	127 (59)		1.76	(0.98–3.15)	
M1	474	265 (56)	0.008	1.01	(0.63–1.63)	0.055

^a^ Chi-square test. ^b^ M0 = non-metastatic, M1 = metastatic. ^c^ Logistic multivariate regression model, AOR = Adjusted odds ratio, *p*: likelihood ratio test.

**Table 4 cancers-14-01675-t004:** Occurrence of severe toxicity leading to a change in or interruption of chemotherapy regimen among patients receiving chemotherapy for pancreatic cancer (*n* = 528).

		Severe Toxicity		Multivariate Analysis ^c^		
	Total	*n* (%)	*p* ^a^	AOR	(95%CI)	*p*
Age						
<65 years	173	51 (29)		1		
65–74 years	204	58 (28)		1.17	(0.73–1.87)	
75–84 years	139	52 (37)		2.59	(1.44–4.63)	
≥85 years	12	8 (67)	0.017	12.40	(3.05–50.44)	<0.001
Sex						
Men	292	81 (28)		1		
Women	236	88 (37)	0.019	1.42	(0.96–2.12)	0.082
Charlson index						
0–1	409	141 (34)		-		
2	81	21 (26)		-		
3+	36	7 (19)	0.077	-		
Performance Status						
0–1	440	141 (32)		-		
2	64	23 (36)		-		
3–4	8	2 (25)	0.744	-		
Deprivation						
Quintile 1	105	30 (29)		-		
Quintile 2	119	32 (27)		-		
Quintile 3	88	31 (35)		-		
Quintile 4	124	42 (34)		-		
Quintile 5	83	33 (40)	0.303	-		
1^st^ line regimen ^b^						
Gem alone	92	24 (26)		1		
Gem_Ox/Gem_Abra alone	78	26 (33)		2.24	(1.07–4.68)	
FFX alone or induction	269	107 (40)		4.31	(2.17–8.53)	
Gem adjuvant	87	12 (14)	<0.001	0.92	(0.39–2.15)	<0.001

^a^ Chi-square test. ^b^ Gem alone = gemcitabine +/- oral capecitabine without resection. Gem_Ox/Gem_Abra alone = gemcitabine + oxaliplatin or gemcitabine + nab-paclitaxel without surgical resection. FFX alone = leucovorine + 5-fluorouracil + irinotecan + oxaliplatin (FOLFIRINOX) without surgical resection. FFX induction = leucovorine + 5-fluorouracil + irinotecan + oxaliplatin (FOLFIRINOX) followed by surgical resection. Gem adjuvant = gemcitabine after surgical resection. ^c^ Logistic multivariate regression model, AOR = Adjusted odds ratio, *p*: likelihood ratio test.

**Table 5 cancers-14-01675-t005:** Net survival (% and 95% Confidence Intervals) at 6 months, 1 year, and 3 years in patients with pancreatic cancer (*n* = 912). Uni and multivariate analysis ^a^.

	6 Months	1 Year	3 Years	*p*	AHR (95% CI)	*p*
Age						
<65 years	69 (62–74)	47 (40–53)	12 (8–17)		1	
65–74 years	70 (64–75)	53 (46–59)	17 (13–22)		0.9 (0.74–1.09)	
75–84 years	48 (42–54)	30 (24–36)	10 (7–15)		1.31 (1.07–1.61)	
≥85 years	27 (20–34)	12 (7–18)	-	<0.001	2.19 (1.70–2.81)	<0.001
Sex						
Men	58 (53–62)	40 (36–45)	13 (10–16)		1	
Women	53 (48–58)	34 (29–39)	9 (6–12)	0.065	0.97 (0.84–1.13)	0.706
Charlson index						
0–1	59 (55–63)	40 (36–43)	10 (8–13)		1	
2	48 (40–56)	36 (28–44)	12 (7–19)		1.06 (0.87–1.30)	
3+	44 (34–54)	26 (18–35)	9 (4–16)	0.029	1.28 (1.00–1.63)	0.166
Performance Status						
0–1	71 (67–74)	50 (45–54)	16 (13–19)		1	
2	36 (29–43)	18 (13–24)	2 (1–6)		1.7 (1.41–2.06)	
3–4	16 (8–25)	9 (4–17)	0 -	<0.001	2.2 (1.65–2.92)	<0.001
Deprivation						
Quintile 1	58 (50–65)	38 (31–46)	16 (10–22)		1	
Quintile 2	66 (58–72)	49 (41–56)	11 (7–17)		0.96 (0.76–1.21)	
Quintile 3	52 (44–60)	35 (27–42)	6 (3–11)		1.09 (0.86–1.38)	
Quintile 4	53 (47–60)	36 (30–43)	14 (9–19)		1.16 (0.93–1.45)	
Quintile 5	48 (40–55)	28 (21–35)	5 (2–10)	0.01	1.06 (0.84–1.33)	0.465
Clinical features ^b^						
M0 resectable tumor	84 (77–88)	69 (62–76)	35 (28–42)		1	
Locally advanced	66 (59–72)	48 (41–55)	8 (5–13)		1.43 (1.13–1.80)	
M1	41 (37–46)	21 (17–25)	3 (2–5)	<0.001	3.31 (2.69–4.09)	<0.001

^a^ Net survival regression model, AHR = adjusted hazard ratio, *p*: likelihood ratio test. ^b^ M0 = non-metastatic, M1 = metastatic.

## Data Availability

On request to corresponding author.

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
