# Peer review of "Management and Outcomes of Pancreatic Cancer in French Real-World Clinical Practice"

_cancers, 2022, doi:10.3390/cancers14071675_

Round 1

Reviewer 1 Report

Jooste et al are describing in a population based study in France the management and outcomes of pancreatic cancer. The study is based on data from their local cancer registry that covers around 1 million habitants. From there the study analyzed data of all malignant exocrine pancreatic neoplasms (n=912). The analysis has been done throughly and the statistical approach is correct as far as I can tell. However, it would have been helpful to understand a bit about the environment of this specific area as it is probably not comparable to large urban areas such as Paris or other European countries. Do they have access to surgical centers? Radiation? Are maybe those infrastructures limiting? My understanding of the study is that decisions are based on age and ECOG or Charlson index but not specific diseases either. Thus, while this certainly of interest, I am not able to draw conclusions for myself as my area may be very different. In addition, it remains unclear why ony 14% of the patients were resected while a larger percentage was "labeled" as resectable - even though this is comparable to other countries per reference from the authors. Speaking of reference: 16 references for a study showing a real world management, is not a lot. I would like to see at least mentions of more guidelines than the 2015 one from ESMO such as Khorana et al from 2017 of ASCO, NCCN  of 2022 (!!!).

Lastly, the English is quite bumpy and would definitely benefit from improvements potentially with help of a scientific writer.

Author Response

We thank the reviewer for his/her relevant and useful comments. We hope that the modifications brought to the manuscript and the point by point responses will be satisfactory.

Jooste et al are describing in a population based study in France the management and outcomes of pancreatic cancer. The study is based on data from their local cancer registry that covers around 1 million habitants. From there the study analyzed data of all malignant exocrine pancreatic neoplasms (n=912). The analysis has been done throughly and the statistical approach is correct as far as I can tell. However, it would have been helpful to understand a bit about the environment of this specific area as it is probably not comparable to large urban areas such as Paris or other European countries. Do they have access to surgical centers? Radiation? Are maybe those infrastructures limiting?

In the limits section of the discussion, the paragraph describing the generalizability was indeed not informative enough. We modified it to include information regarding the local environment:

“The generalizability of our findings, particularly with respect to clinical practices, de-pends as much on the system of care as on the modalities of its use, which may vary with the social characteristics of patients [22]. However, the supply of healthcare in the studied region showed similar characteristics to those of other French geographic areas (reference academic cancer centers, private and public hospitals). Access to radiotherapy, surgery or chemotherapy, in both private and public structures was un-limited, and the rules for authorizing the activities of healthcare facilities have been defined for the whole of France by the national Cancer Plan (https://www.e-cancer.fr/Professionnels-de-sante/L-organisation-de-l-offre-de-soins/Traitements-du-cancer-les-etablissements-autorises/Les-autorisations-de-traitement-du-cancer). French patients have the same access to national health insurance-funded care and to any place of treatment, whatever its geographic situation.”

My understanding of the study is that decisions are based on age and ECOG or Charlson index but not specific diseases either. Thus, while this certainly of interest, I am not able to draw conclusions for myself as my area may be very different. In addition, it remains unclear why ony 14% of the patients were resected while a larger percentage was "labeled" as resectable - even though this is comparable to other countries per reference from the authors. Speaking of reference: 16 references for a study showing a real world management, is not a lot. I would like to see at least mentions of more guidelines than the 2015 one from ESMO such as Khorana et al from 2017 of ASCO, NCCN of 2022 (!!!).

We have added 6 references (highlighted in the manuscript). As this study used exhaustive population-based registry data to describe real-world management, we chose to compare our results with those of other, mainly European, cancer registry studies. Literature on the management of pancreatic cancer is scarce. To our knowledge, there have been no population-based studies on resection rates among resectable tumors. Regarding guidelines, we opted to refer only to the 2015 edition, which was in use during the management of patients included in this study (2014-2017). We have now added the 2019 ASCO guidelines by Khorana et al in the discussion.

Lastly, the English is quite bumpy and would definitely benefit from improvements potentially with help of a scientific writer.

The English has been revised by a native scientific writer.

Reviewer 2 Report

This is a study using the registry, so I think it is very meaningful.

  • I don’t understand what the EDI deprivation index indicates. Please describe the EDI deprivation index in more detail.
  • In Table 1, please describe the percentage of patients who received best supportive care without surgery or chemotherapy.
  • In figure 2, it is possible that surgery was performed on selected patients with good condition in the 75-84 age group. More elderly patients could have received surgery or chemotherapy, however, the Charlson index was not associated with treatment or survival. Please describe what approach you think is best.  
  • Line 157, EDI is an abbreviation and should be spelled out. 

Author Response

We thank the reviewer for his/her relevant and useful comments. We hope that the modifications brought to the manuscript and the point by point responses will be satisfactory.

This is a study using the registry, so I think it is very meaningful.

  • I don’t understand what the EDI deprivation index indicates. Please describe the EDI deprivation index in more detail.

The Data set section has been modified, spelling out EDI, and 1 reference has been added:

“Socio-economic status was assessed at the level of the “IRIS”, which is the smallest geographic areas defined by the ‘Institut National de la Statistique et des Etudes Economiques’ for which census data are available[6]. Each IRIS includes approximately 2000 individuals with relatively homogeneous social characteristics. An IRIS was assigned to each patient according to their residence address at the time of diagnosis. The French version of the ecological European Deprivation Index (EDI), an aggregate index of deprivation based on the 2011 national census, was used to assign an EDI score to each IRIS [7]. The continuous EDI was categorized into national quintiles (the higher the quintile, the greater the social deprivation). The EDI was missing for 13 cases.”

  • In Table 1, please describe the percentage of patients who received best supportive care without surgery or chemotherapy.

Table 1 has been modified to describe the percentage of patients who received best supportive care.

  • In figure 2, it is possible that surgery was performed on selected patients with good condition in the 75-84 age group. More elderly patients could have received surgery or chemotherapy, however, the Charlson index was not associated with treatment or survival. Please describe what approach you think is best.  

It is indeed very probable that selection for surgery was more drastic in the 75-84 age group than in younger patients. We also think that in epidemiological studies, information on performance status seems to better reflect clinical operability and patients’ ability to withstand burdensome treatment than does the comorbidities index alone.

  • Line 157, EDI is an abbreviation and should be spelled out. 

The Data set section has been modified: EDI has been spelled out - European Deprivation Index.

Reviewer 3 Report

This is a population based study of clinical outcomes of pancreatic cancer.

  1. I am not familiar with this registry but how was informed consent was waived. Please add in the text.
  2.  Why was gem+capecitabine considered as gem alone?
  3. Patients who received FFX as neoadjuvant chemotherapy but could not undergo surgical resection would be included in induction in clinical practice. Can those information on patients who underwent neoadjuvant treatment with curative resection intent?
  4. How were data on RECIST collected, and who evaluated RECIST? 
  5. Please add number at risk in Figures showing Kaplan-Meier analyses.

Author Response

We thank the reviewer for his/her relevant and useful comments. We hope that the modifications brought to the manuscript and the point by point responses will be satisfactory.

This is a population based study of clinical outcomes of pancreatic cancer.

  1. I am not familiar with this registry but how was informed consent was waived. Please add in the text.

In agreement with the French legislation, there is no requirement for written informed consent for validated public cancer registries. The following sentence was added in the Patients section: “This observational non-interventional study was approved by the French Data Protection Authority (CNIL, authorization n° 998024), and in agreement with French legislation, there was no requirement for written informed consent.”

  1. Why was gem+capecitabine considered as gem alone?

We thank the reviewer for pointing out a mistake: Gemcitabine without resection was always given without capecitabine, whereas in adjuvant settings, gemcitabine was given alone or with capecitabine.

We pooled gemcitabine+capecitabine with gemcitabine following the ASCO guidelines (Khorana et al JCO 2019), which state that doublet therapy with gemcitabine and capecitabine or mono-therapy with gemcitabine alone can be offered in adjuvant settings. They followed the phase III ESPAC-4 trial (Neoptolemos et al Lancet. 2017) comparing the two regimens showed no difference in disease-free survival, and the addition of capecitabine to gemcitabine was achieved with manageable toxicity and no detrimental effect on quality of life.

The Methods section has been modified as follows:

“0) No chemotherapy (N=375)

1) gemcitabine without resection (Gem alone, N=92),

2) gemcitabine+oxaliplatin or gemcitabine+nab-paclitaxel without surgical resec-tion (Gem_Ox/Gem_Abra alone, N=78); these were pooled as patients presented simi-lar clinical features (mostly locally advanced or M1 disease) and similar characteris-tics,

3) leucovorine+5-fluorouracil+irinotecan+oxaliplatin without surgical resection (FOLFIRINOX: FFX alone, N=255),

4) FOLFIRINOX followed by surgical resection (FFX induction, N=14),

5) gemcitabine (N=63) or gemcitabine+capecitabine (N=24) after surgical resec-tion (Gem adjuvant, N=87). In accordance with ASCO guidelines [9] stating that dou-blet therapy with gemcitabine and capecitabine or mono-therapy with gemcitabine alone can be offered in adjuvant settings; gemcitabine+capecitabine and gemcitabine alone were pooled.”

  1. Patients who received FFX as neoadjuvant chemotherapy but could not undergo surgical resection would be included in induction in clinical practice. Can those information on patients who underwent neoadjuvant treatment with curative resection intent?

Population-based observational studies led by cancer registries are different from clinical trials. It is not feasible to collect the intention to treat retrospectively and exhaustively in medical files. We have modified the methods section and the tables to improve clarity: FFX before surgical resection (FFX induction, N=14), has been replaced by FFX followed by surgical resection (FFX induction, N=14).

  1. How were data on RECIST collected, and who evaluated RECIST? 

RECIST was collected retrospectively from each medical file through the special survey set up to complete routinely registered data in the registry. RECIST was mostly notified in the iterative multidisciplinary team meetings or in practitioners’ reports. Thus, the registry staff collected the iterative RECIST conclusions evaluated by the patients’ practitioners. The Dataset section was modified as follows to be clearer:

“Progression was assessed according to the RECIST (Response Evaluation Criteria in Solid Tumors) definition. The registry staff retrospectively collected and dated the iterative RECIST conclusions evaluated by the patients’ practitioners. Progression was available in 91% of patients with non-metastatic cancer who underwent surgical re-section or first-line chemotherapy.”

  1. Please add number at risk in Figures showing Kaplan-Meier analyses.

Figures have been modified.

Round 2

Reviewer 1 Report

Suggestions have been addressed correctly.